# An Efficient SAR Raw Signal Simulator Accounting for Large Trajectory Deviation

**DOI:** 10.3390/s25144260

**Published:** 2025-07-09

**Authors:** Shaoqi Dai, Haiyan Zhang, Cheng Wang, Zhongwei Lin, Yi Zhang, Jinhe Ran

**Affiliations:** College of Electronics and Information, National University of Defense Technology, Hefei 230031, China; daishaoqi@nudt.edu.cn (S.D.); chengwang@nudt.edu.cn (C.W.);

**Keywords:** synthetic aperture radar (SAR), raw signal, simulation, large trajectory deviation

## Abstract

A synthetic aperture radar (SAR) raw signal simulator is useful for supporting algorithm innovation, system scheme verification, etc. Trajectory deviation is a realistic factor that should be considered in a SAR raw signal simulator and is very important for applications such as motion composition and image formation for a SAR with nonlinear trajectory. However, existing efficient simulators become deteriorated and even invalid when the magnitude of trajectory deviation increases. Therefore, we designed an efficient SAR raw signal simulator that accounts for large trajectory deviation. Based on spatial spectrum analysis of the SAR raw signal, it is disclosed and verified that the 2D spatial frequency spectrum of the SAR raw signal is an arc of a circle at a fixed transmitted signal frequency. Based on this finding, the proposed method calculates the SAR raw signal by curvilinear integral in the 2D frequency domain. Compared with existing methods, it can precisely simulate the SAR raw signal in the case that the deviation radius is much larger. Moreover, taking advantage of the fast Fourier transform (FFT), the computational complexity of this method is much less than the time-domain ones. Furthermore, this method is applicable for multiple SAR acquisition modes and diverse waveforms and compatible with radar antenna beam width, squint angle, radar signal bandwidth, and trajectory fluctuation. Experimental results show its outstanding performance for simulating the raw signal of SAR with large trajectory deviation.

## 1. Introduction

Synthetic aperture radar (SAR) raw data simulation is very useful for evaluating algorithms, verifying system design, planning SAR missions, etc. [1]. After years of development, SAR signal simulators are being applied to increasingly more domains, including urban structures [2], ocean environments [3,4], and forest settings [5]. Furthermore, raw signal simulators have wide applicability to increasingly more SAR systems. On the one hand, SAR simulators have been designed not only for the classic stripmap mode, but also for multiple radar acquisition modes [6,7], such as sliding spotlight SAR [8] and TopSAR [9]. On the other hand, the application of raw signal simulation has been expanded from traditional monostatic radar configuration to bistatic SAR [10], and from pulsed SAR to frequency modulation continuous wave SAR [11,12].

As the most straightforward approach, the time-domain simulator is the most accurate and flexible simulator. Using the superposition principle, it calculates a coherent sum of the target echo for each transmitted radar impulse. Thus, time-domain simulators can have not only high precision but also strong generality. This approach can easily consider the real orbit of the platform and other effects such as mechanical structure oscillation and target backscattering coefficient variation during the system integration time. Moreover, it has the least restrictive conditions. It almost has no constraint on the scene and radar parameters or acquisition mode. Unfortunately, for extended scenes, it is enormously time-consuming [13], which limits its applications. Although advancements in hardware performance have mitigated the urgency of computational efficiency concerns, the exponential growth in data volume driven by enhanced radar imaging precision continues to impose stringent requirements on algorithmic complexity. Furthermore, in specialized applications such as SAR deception jamming [14,15], the demand for real-time processing necessitates the ongoing development of efficient simulation algorithms.

In order to improve the calculation efficiency, many Fourier-domain simulators have been proposed in the case of an ideal straight-line trajectory [16]. By utilizing the efficient processing of fast Fourier transforms (FFTs), the computational complexity of Fourier-domain simulators becomes much less than that of time-domain simulators. Compared with the time-domain simulator, the flexibility of the Fourier-domain simulator is poorer. It inevitably has some constraints on the scene and radar trajectory.

A simulator for SAR with trajectory deviation is particularly important for certain applications. In order to make the simulator more realistic, trajectory deviation is an important condition to be considered, because the existence of platform deviations is inevitable for airborne and unmanned aerial vehicle SAR. In addition, the SAR raw signal simulators that consider motion errors are important to verify the effectiveness of motion compensation algorithms. They facilitate the analysis of the validity of these algorithms for diverse trajectories, mission parameter choices, and different kinds of imaged scenes. Furthermore, simulators accounting for large trajectory deviation are very useful for testing certain SAR imaging algorithms, such as the study of SAR imaging under the condition of nonlinear trajectory [17,18].

After years of study, simulators that consider trajectory deviation have been developed. However, the existing simulators have many constraints on radar parameters, acquisition model, or/and trajectory. For example, the method presented in [19] is under the strict constraint that the range and azimuth beam of radar antenna are narrow and the trajectory deviation is sufficiently slow. Although the assumption of narrow beam and slow deviation is relaxed in [20], this relaxation is at the expense of computational efficiency. The frequency-domain simulator in [21] can be implemented only when the bandwidth of the radar signal is narrow. Furthermore, these methods are only applicable for stripmap mode with zero squint angle and are subject to the magnitude of trajectory deviation. In contrast, the simulator proposed in [22] based on spatial spectrum analysis eliminates the constraints on radar parameters and acquisition model. However, the computational efficiency is offset by the stringent constraints of the trajectory deviation radius. Generally, the existing efficient SAR simulators cannot operate well under the condition of large trajectory deviation.

This paper introduces a novel SAR raw signal simulator that accommodates large trajectory deviations, leveraging spatial spectrum analysis. The method computes the raw signal via a curvilinear integral in the 2D spatial frequency domain, grounded in the principle that the 2D spatial frequency spectrum of the SAR signal forms an arc of a circle at a fixed frequency. The core operations of the proposed method include 2D FFT, Stolt interpolation, antenna pattern modulation, spectrum correction, and integration.

The primary contributions of this work are as follows:

First, the proposed method effectively handles large trajectory deviations, addressing the limitations of conventional Fourier-domain methods, which are constrained by the magnitude of trajectory deviations. As the deviation radius increases, these methods suffer from reduced accuracy and efficiency, often becoming inapplicable. In contrast, the spatial spectrum-based approach directly models nonlinear trajectories by interpreting the SAR raw signal as a spatial function of radar location at a fixed-range frequency, thereby mitigating the constraints on deviation magnitude.

Second, the method significantly enhances computational efficiency compared to time-domain simulators, which are computationally intensive, especially for large-scale scenes. By transforming the integral into a 2D Fourier transform, which needs to be calculated only once, the method reduces computational complexity. Notably, as the deviation radius increases, the computational complexity remains largely unaffected.

Third, the proposed simulator exhibits wide applicability and strong generality. It supports various SAR acquisition modes and waveform models. By defining a rational transfer function, the method accommodates different SAR acquisition scenarios, including variations in radar antenna beamwidth, squint angle, signal bandwidth, and trajectory fluctuations, all while easing the constraints on the waveform.

The structure of this paper is as follows: Section 2 outlines the SAR system geometry and presents the spatial domain perspective of the SAR raw signal. Section 3 derives the mathematical framework of the proposed method and details the processing procedures. Section 4 evaluates the performance of the simulator, with experimental validation provided in Section 5. Finally, conclusions are presented in Section 6.

## 2. Model of the Proposed Method

### 2.1. Geometry of SAR System with Trajectory Deviation

Figure 1a illustrates the geometry of a SAR system with trajectory deviation in a three-dimensional coordinate system. The nominal trajectory aligns with the coordinate axes. Typically, azimuthal deviations are corrected via azimuth resampling of SAR raw data [23], so this study focuses solely on cross-track deviations. Key notations are as follows:
(u,v) the azimuth and (slant) range coordinates of the scene generic scattering point *P*;(x,r) the azimuth and (slant) range coordinates of the antenna;$\varphi \left(x\right)$ pitch angle of the platform;*R* the target-to-antenna distances in the generic antenna position for actual trajectories;$R_{0}$ the closest range to the nominal trajectory (in nominal trajectory, $R_{0}=v$).

The antenna’s position is expressed using azimuth coordinate *x* and deviation vector $\vec{d}$, with $d_{y}\left(x\right)$ and $d_{z}\left(x\right)$ representing horizontal and vertical deviations, respectively. The instantaneous slant range *R* is given as:(1)$$R=\sqrt{{\left(R_{0}+\Delta R_{0}\right)}^{2}+{\left(u-x\right)}^{2}}$$
where $\Delta R_{0}$ is derived via Taylor’s theorem (see Figure 1b):(2)$$\begin{array}{cc} \Delta R_{0} & =\sqrt{R_{0}^{2}+{d\left(x\right)}^{2}-2d\left(x\right)R_{0}\cos\theta_{⊥}}-R_{0}\\ & \approx -d\left(x\right)\cos\theta_{⊥}+\frac{d{\left(x\right)}^{2}sin^{2}\theta_{⊥}}{2R_{0}}+\frac{d{\left(x\right)}^{3}cos\theta_{⊥}sin^{2}\theta_{⊥}}{2{R_{0}}^{2}} \end{array}$$

Here, $d\left(x\right)=∥\vec{d(x)}∥$, $\theta_{⊥}=\varphi \left(x\right)+\beta \left(x\right)$, and β(x) depends on platform deviations. The approximation holds for small deviations relative to the target’s slant range. In the slant plane, *r* is defined as(3)$$\begin{array}{c} r=-\Delta R_{0}=d\left(x\right)\cos\theta_{⊥}-\frac{d{\left(x\right)}^{2}sin^{2}\theta_{⊥}}{2R_{0}}-\frac{d{\left(x\right)}^{3}cos\theta_{⊥}sin^{2}\theta_{⊥}}{2{R_{0}}^{2}} \end{array}$$

Thus, $R_{0}=v$, and *R* simplifies to(4)$$R=\sqrt{{\left(u-x\right)}^{2}+{\left(v-r\right)}^{2}}$$

This establishes a two-dimensional geometric model for SAR simulation in the slant plane.

### 2.2. Model of SAR Raw Signal

The Fourier transform of the received radar echo in the fast-time domain, denoted as $S_{R}\left({\vec{l}}_{r};k\right)$, is given by(5)$$S_{R}\left({\vec{l}}_{r};k\right)=S_{T}\left({\vec{l}}_{r};k\right)hh_{tran}\left({\vec{l}}_{r};k\right)$$
where:${\vec{l}}_{r}=\left(x,r\right)$ is the radar’s instantaneous location;$k=2f_{r}/c$, with $f_{r}$ as frequency and *c* as the wave propagation speed;$S_{T}\left({\vec{l}}_{r};k\right)$ is the transmitted radar signal;$hh_{\mathrm{tran}}\left({\vec{l}}_{r};k\right)$ is the SAR transfer function:(6)$$hh_{tran}\left({\vec{l}}_{r};k\right)=rect\left(\frac{k-k_{0}}{B}\right)\;\cdot \;\int \int \;hh_{tran}\left({\vec{l}}_{r};{\vec{l}}_{s};k\right)dudv.$$

Here, ${\vec{l}}_{s}=\left(u,v\right)$ denotes a scatterer’s location, rect(·) is a rectangular window centered at $k_{0}$ with bandwidth *B*, and(7)$$hh_{tran}\left({\vec{l}}_{r};{\vec{l}}_{s};k\right)=s_{amp}\left({\vec{l}}_{r};{\vec{l}}_{s}\right)\exp\left(-j2\pi kR\left({\vec{l}}_{r};{\vec{l}}_{s}\right)\right)$$

The amplitude attenuation $s_{\mathrm{amp}}\left(\cdot \right)$ is expressed as(8)$$s_{amp}\left({\vec{l}}_{r};{\vec{l}}_{s}\right)=\frac{\omega \left(\theta \left({\vec{l}}_{r};{\vec{l}}_{s}\right),\theta_{0}\left({\vec{l}}_{r}\right)\right)\sigma \left({\vec{l}}_{s}\right)}{{\left(4\pi \right)}^{3/2}R{\left({\vec{l}}_{r};{\vec{l}}_{s}\right)}^{2}}$$
where $\omega \left(\cdot \right)$ is the amplitude induced by the beam pattern of the radar antenna; θl→r;l→s=∧l→s−l→rl→s−l→rl→s−l→r/l→s−l→r is the instantaneous direction of the scatterer with respect to the radar antenna; $∥\cdot ∥$ denotes the norm; $\theta_{0}\left({\vec{l}}_{r}\right)$ represents the pointing direction of the radar main lobe; $\sigma \left(\cdot \right)$ is the backscattering coefficient of the scatterer; and $R\left({\vec{l}}_{r};{\vec{l}}_{s}\right)\overset{\wedge }{=}∥{\vec{l}}_{s}-{\vec{l}}_{r}∥$ is the distance between the scattering point and antenna.

This spatial-domain SAR signal model differs from conventional approaches in three ways:
Spatial Representation: the SAR raw signal is viewed as a function of radar location ${\vec{l}}_{r}$ rather than the conventional fast time (slant range) and slow time (azimuth). This is particularly suitable for handling nonlinear radar trajectories.Applicability to Multiple Modes: the model applies to various SAR acquisition modes. For instance, a constant θ corresponds to stripmap mode, while variable θ enables modes like sliding spotlight or TOPSAR (see Figure 2).Factorized Structure: the SAR signal is split into scatterer-independent $S_{T}\left(\cdot \right)$ and scatterer-dependent $hh_{\mathrm{tran}}\left(\cdot \right)$. Efficient computation of $hh_{\mathrm{tran}}\left(\cdot \right)$ is key to simulator design.

## 3. Theory of the Proposed Method

### 3.1. Mathematical Principle

According to (Equation 6), the transfer function $hh_{tran}\left({\vec{l}}_{r};k\right)$ can be calculated using a double integral. However, notably, $hh_{tran}\left(\cdot \right)$ is a function of antenna location ${\vec{l}}_{r}$ and signal frequency *k*. Moreover, the double integral varies with different ${\vec{l}}_{r}$ and *k*. In order to obtain the raw signal, we must calculate the double integral at each ${\vec{l}}_{r}$ and *k*. It is evident that this approach is not efficient.

To improve computational efficiency, we analyze the structure of the transfer function by transforming Equation (Equation 6) into the two-dimensional frequency domain. Through this process, we observe that the original two-dimensional spatial integral over the reflectivity map σ(·) can be interpreted as the two-dimensional Fourier transform of σ(·). More importantly, we find that the spatial spectrum of the transfer function $hh_{\mathrm{tran}}\left({\vec{l}}_{r};k\right)$ lies along a curved trajectory in the two-dimensional $\left(k_{x},k_{r}\right)$ frequency space. This key observation enables a reformulation of the transfer function as a one-dimensional integral along this curved path, rather than evaluating a two-dimensional integral for each antenna position ${\vec{l}}_{r}$ and frequency *k*. Since both ${\vec{l}}_{r}$ and *k* vary during signal acquisition, avoiding the double integral greatly reduces computational load. (The detailed derivation is provided in Section A.1 and Section A.2).

Accordingly, the transfer function can be expressed as(9)$$\begin{array}{cc} hh_{tran}\left(x,r;k\right)= & \int_{L}\;H_{bw}\left(k_{x},k_{r}\right)\\ & \cdot C\left(k_{x},k_{r},r\right)\cdot \Gamma \left(k_{x},k_{r}\right)\cdot e^{j2\pi k_{x}x}dk_{x} \end{array}$$

This equation indicates that the transfer function can be calculated using a curvilinear integral, whose essence is the inverse Fourier transform operation. The exponential term $e^{j2\pi k_{x}x}$ is the integration factor of the inverse Fourier transform. The details of each term are described as follows:(1)*L* is the arc on circle ${k_{x}}^{2}+{k_{r}}^{2}=k^{2}$, which satisfies(10)$$\theta \left(k_{x},k_{r}\right)\in \left[\theta_{sq}-\theta_{bw}/2,\theta_{sq}+\theta_{bw}/2\right]$$
where $\theta \left(k_{x},k_{r}\right)=tan^{-1}\left(k_{x}/k_{r}\right)$ denotes the angle in the Cartesian coordinates $\left(k_{x},k_{r}\right)$. This integral interval is because the 2D spatial frequency spectrum of the transfer function $hh_{tran}\left(\cdot \right)$ is band-limited, as illustrated in Figure 3. Note that the integral is along the direction of $k_{x}$ growth.(2)$H_{bw}\left(\cdot \right)$ is relevant to the bandwidth and beam pattern, and can be expressed as follows:(11)$$H_{bw}\left(k_{x},k_{r}\right)=rect\left(\frac{\sqrt{{k_{x}}^{2}+{k_{r}}^{2}}-k_{0}}{B}\right)\cdot \omega \left[\theta \left(k_{x},k_{r}\right)-\theta_{sq}\right]$$The rectangular window $rect\left(\cdot \right)$ is obtained from the fact that the bandwidth of the transmitted signal is limited. As shown in Figure 3, the radius $\sqrt{{k_{x}}^{2}+{k_{r}}^{2}}$ in $\left(k_{x},k_{r}\right)$ domain represents the wavenumber *k*. According to $rect\left(\cdot \right)$, the bandwidth is *B* with the carrier frequency $k_{0}$. Furthermore, the 2D spatial frequency spectrum of $hh_{tran}\left(\cdot \right)$ depends on the beam pattern of the radar beam. As illustrated by the window function $\omega \left(\cdot \right)$, the spectrum of transfer function is modulated by the beam pattern $\omega \left(\cdot \right)$.(3)$C\left(k_{x},k_{r},r\right)$ is the correction term(12)$$C\left(k_{x},k_{r},r\right)=C_{0}\cdot {\left(k_{x}^{2}+k_{r}^{2}\right)}^{-\frac{5}{4}}\cdot k_{r}^{2}\cdot e^{j2\pi k_{r}r}$$
where $C_{0}$ is a constant irrelevant to this work. Further, the exponential term $e^{j2\pi \sqrt{k^{2}-k_{x}^{2}}r}$ is obtained from the trajectory deviation.(4)$\Gamma \left(k_{x},k_{r}\right)$ is the 2D Fourier transform of $\sigma \left(u,v\right)$ weighted by $v^{-\frac{3}{2}}$.(13)$$\begin{array}{c} \Gamma \left(k_{x},k_{r}\right)\;=\;\int \int \;\sigma \left(u,v\right)\cdot v^{-\frac{3}{2}}\cdot e^{-j2\pi k_{r}v}e^{-j2\pi k_{x}u}dudv \end{array}$$

The reason why the transfer function can be obtained using a curvilinear integral, rather than iterative double integrals, is related to the 2D spatial frequency spectrum of the transfer function. The spectrum satisfies(14)$${k^{2}=k_{x}}^{2}+{k_{r}}^{2},tan^{-1}\left(\frac{k_{x}}{k_{y}}\right)\in \left[\theta_{sq}-\frac{\theta_{bw}}{2},\theta_{sq}+\frac{\theta_{bw}}{2}\right]$$

In other words, the 2D spatial frequency spectrum of SAR transfer function would be an arc of a circle, as shown in Figure 3. Thus, the essence of our method is transforming the double integral to a 2D Fourier transform and calculating the transfer function using a curvilinear integral in the 2D frequency domain. Notably, the 2D Fourier transform $\Gamma \left(k_{x},k_{r}\right)$ is independent of the antenna location ${\vec{l}}_{r}$ and wavenumber *k*. Thus, the 2D Fourier transform can be applied only once, which reduces the complexity of this algorithm significantly in comparison with the double integral method.

### 3.2. Implementation

In this paper, we propose an efficient SAR raw signal simulation approach. The procedure is illustrated by the block scheme in Figure 4. From the block scheme, we observe that the procedures can be divided into two parts: the calculation of transfer function $hh_{tran}\left(\cdot \right)$ and the computation of the SAR raw signal.

The vertical procedure is employed to obtain the transfer function $hh_{tran}\left({\vec{l}}_{r};k\right)$, which includes five steps. Each step is specified as follows:

Step (1) 2D FFT of weighted reflectivity map. As expressed in (Equation 13), the object of the 2D Fourier transform is the weighted backscatter coefficient. Thus, $\sigma \left(u,v\right)$ should be modulated according to(15)$$\gamma \left(u,v\right)=\sigma \left(u,v\right)\cdot v^{-\frac{3}{2}}$$

Generally, the modulated reflectivity pattern $\gamma \left(u,v\right)$ is modeled as an impulse train in 2D continuous space domain(16)$$\begin{array}{c} \gamma \left(u,v\right)=\sum_{m}\sum_{n}\gamma \left[m,n\right]\delta \left(u-m\Delta u,v-n\Delta v\right) \end{array}$$
where $\delta \left(\cdot ,\cdot \right)$ is a 2D Dirac function; Δu,Δv are the spatial sampling intervals in range and azimuth, respectively. Further, $\gamma \left[m,n\right]$ is a discrete sequence that denotes the weighted backscattering coefficient at $\left(m\Delta u,n\Delta v\right)$: (17)$$\gamma \left[m,n\right]=\gamma \left(u,v\right){|}_{u=m\Delta u,v=\Delta v}$$

Subsequently, the 2D FFT operation is applied to $\gamma \left[m,n\right]$ and the samples of $\Gamma \left(k_{r},k_{x}\right)$ are obtained on a uniform rectangular grid of the coordinates $\left(k_{r},k_{x}\right)$:(18)$$\begin{array}{c} \mathrm{FFT}\left\{\gamma \left[m,n\right]\right\}=\Gamma \left(k_{x},k_{r}\right){|}_{k_{x}=l_{x}\Delta k_{x},k_{r}=l_{r}\Delta k_{r}}\\ \;\;\;\;\;\;\;\;\Delta k_{x}=1/\left(N_{u}\Delta u\right),\;l_{x}=0,1,...,N_{u}-1\\ \;\;\;\;\;\;\;\;\Delta k_{r}=1/\left(N_{v}\Delta v\right),\;l_{r}=0,1,...,N_{v}-1 \end{array}$$
where $\Delta k_{x}$ and $\Delta k_{r}$ are the sampling intervals of *x*-frequency and *r*-frequency, respectively, and $N_{u}$ and $N_{v}$ are FFT lengths. According to the theory of digital signal processing, $\Gamma \left(k_{r},k_{x}\right)$ is periodical with the periods 1/Δu and 1/Δv. Therefore, after performing the 2D FFT of $\gamma \left(u,v\right)$, we can obtain the samples of $\Gamma \left(k_{x},k_{r}\right)$ in the entire 2D frequency plane $\left(k_{x},k_{r}\right)$.

Step (2) Spectrum extraction. The bandwidths of the radar signal and the spatial frequency of the transfer function are both limited. Thus, this step is to extract the frequency spectrum from the entire 2D frequency plane $\left(k_{x},k_{r}\right)$, as shown in Figure 5a. The portion to be extracted is a part of the circular annulus, which is referred to as the support region of radar. Its radial segment is centered at the carrier frequency of radar $k_{0}$, and the radial extent represents the bandwidth of the radar signal *B*. A part of the annulus is tilted by the amount of squint angle $\theta_{sq}$ of radar, and the angular extent is the antenna beam width $\theta_{bw}$.

Note that, if the acquisition mode is not stripmap, the squint angle $\theta_{sq}$ varies with the instantaneous antenna position ${\vec{l}}_{r}$. In other words, the support region rotates around the origin with ${\vec{l}}_{r}$ due to the variation of $\theta_{sq}$. Thus, in order to obtain the support region for the entire trajectory, the angular extent should be added ${\left[\theta_{sq}\right]}_{bw}$, the varying range of squint angle, $max\left[\theta_{sq}\right]-min\left[\theta_{sq}\right]$. In addition, the angle of the annulus tilted should be the mean of the squint angle.

Step (3) Stolt interpolation. After the first two steps, we obtain $\Gamma \left(k_{x},k_{r}\right)$ within the support region. However, the samples of $\Gamma \left(k_{x},k_{r}\right)$ are on a uniform rectangular grid of the coordinates $\left(k_{x},k_{r}\right)$, as shown in Figure 5a. According to (Equation 9), the transfer function is calculated using a curvilinear integral. Thus, subsequently, we require samples of $\Gamma \left(k_{x},k_{r}\right)$ located on the circular arc within the support region, as shown in Figure 5b. This objective can be achieved by resampling (or interpolating) $\Gamma \left(k_{x},k_{r}\right)$ along *x*-frequency $k_{x}$, which is referred to as Stolt interpolation.

Notably, the Stolt interpolation is performed in the spatial frequency domain. Moreover, the center of scene reflectivity pattern $\sigma \left(u,v\right)$ locates $\left(u_{m},v_{m}\right)$ with the widths $u_{b}$ and $v_{b}$ in the range and azimuth, respectively. Thus, the Stolt interpolation involved is a passband filter with the center frequency $\left(-u_{m},-v_{m}\right)$ and bandwidth $\left(u_{b},v_{b}\right)$.

Step (4) Antenna pattern modulation and spectrum correction. According to (Equation 11), the weight of $\Gamma \left(k_{x},k_{r}\right)$ should be altered corresponding to the azimuth beam pattern of the radar. Thus, the weight $\Gamma \left(k_{x},k_{r}\right)$ with the function $\omega \left(\cdot \right)$. Notably, if the acquisition mode is not stripmap, the weight function $\omega \left(\cdot \right)$ should be adjusted with ${\vec{l}}_{r}$. Furthermore, as expressed in (Equation 12), before the integral, the 2D spectrum $\Gamma \left(k_{x},k_{r}\right)$ demands amplitude and phase correction, which is related to the instantaneous location ${\vec{l}}_{r}$. In other words, both the modulation and correction operation should be repeated at different ${\vec{l}}_{r}$.

Step (5) Curvilinear integral. The transfer function, as given by (Equation 9), can be calculated using a curvilinear integral, which, in essence, is the inverse Fourier transform. Notably, after the integral, we obtained the transfer function $hh_{tran}\left(\cdot \right)$ at the position point ${\vec{l}}_{r}$ and frequency sample *k*. In order to obtain the transfer function for the instantaneous location ${\vec{l}}_{r}$, we must repeat the curvilinear integral for each *k*. Furthermore, $hh_{tran}\left(\cdot \right)$ for the entire trajectory can be obtained by repeating the above operation for each ${\vec{l}}_{r}$.

The purpose of the horizontal procedure is to obtain the echoed signal. According to Equation (Equation 5), the Fourier spectrum of the received radar echo $S_{R}\left(\cdot \right)$ can be calculated by multiplying the transfer function $hh_{tran}\left(\cdot \right)$ and the spectrum of the transmitted signal $S_{T}\left(\cdot \right)$. Through the above operation, $hh_{tran}\left(\cdot \right)$ is calculated. Thus, FFT operation is applied on the transmitted signal along the fast time, and subsequently multiplied with $hh_{tran}\left(\cdot \right)$. Finally, the echoed signal can be obtained using the inverse Fourier transform.

## 4. Performance Analysis of the Method

### 4.1. Computational Complexity

The number of floating point operations (FLOPs) is widely used to measure the computational complexity of an algorithm. An FFT or inverse FFT (IFFT) of length *N* requires $5\log_{2}\left(N\right)$ FLOPs, a complex phase multiplication requires six FLOPs, and interpolation requires $2\left(2L-1\right)$ FLOPs for a single complex output, where *L* is the length of the interpolation filter [24]. Thus, the number of FLOPs involved is illustrated as follows:
Weighting the reflectivity map requires $n_{a}n_{r}$ FLOPs, where $n_{a}$ and $n_{r}$ are the number of scatterers in the azimuth and range directions, respectively. Furthermore, the 2D FFT operation of modulated backscattering coefficient requires $5n_{a}n_{r}\log_{2}\left(n_{a}n_{r}\right)$ FLOPs.Spectrum extraction and interpolation along the azimuth frequency require $2\left(2L^{\prime }-1\right)N_{r}n_{a}$ FLOPs, where $N_{r}$ is the sampling number of SAR raw signal in the range direction and $L^{\prime }$ is the length of the interpolation kernel.Antenna pattern modulation and spectrum correction can be applied together. This operation is a complex multiplication and requires $6N_{r}n_{a}$ FLOPs at the instantaneous location ${\vec{l}}_{r}$. For the entire trajectory, these operations should be repeated for each position and require $6N_{r}N_{a}n_{a}$ FLOPs, where $N_{a}$ is the sampling number of SAR raw signal in the azimuth direction.At the instantaneous location ${\vec{l}}_{r}$, the integration should be repeated $N_{r}$ times for each *k* and thus requires $2N_{r}\left(n_{a}-1\right)$ FLOPs. In order to obtain $hh_{tran}\left(\cdot \right)$ for the entire trajectory, we should apply the integral operation for each ${\vec{l}}_{r}$. Therefore, the integral operation for the entire trajectory requires $2N_{r}N_{a}\left(n_{a}-1\right)$ FLOPs.One-dimensional FFT operation and complex multiplication require $5\log_{2}\left(N_{r}\right)$ FLOPs and $6N_{r}N_{a}$ FLOPs, respectively. Finally, 1D IFFT operation requires $5N_{a}\log_{2}\left(N_{r}\right)$, because this operation should be repeated at each ${\vec{l}}_{r}$.

Thus, the overall computational complexity of the aforementioned algorithm is(19)$$\begin{array}{cc} & \underset{weighting}{\underset{︸}{n_{a}n_{r}}}+\underset{2D\;FFT}{\underset{︸}{5n_{a}n_{r}\log_{2}\left(n_{a}n_{r}\right)}}+\underset{extracting\;and\;interpolation}{\underset{︸}{2\left(2L^{\prime }-1\right)N_{r}n_{a}}}+\underset{modulation\;and\;correction}{\underset{︸}{6N_{r}N_{a}n_{a}}}\\ & +\underset{integral}{\underset{︸}{2N_{r}N_{a}\left(n_{a}-1\right)}}+\underset{FFT}{\underset{︸}{5\log_{2}\left(N_{r}\right)}}+\underset{mutiplication}{\underset{︸}{6N_{r}N_{a}}}+\underset{IFFT}{\underset{︸}{5N_{a}\log_{2}\left(N_{r}\right)}}\;\approx \;o\left(N_{r}N_{a}n_{a}\right) \end{array}$$

Notably, the most time-consuming operations are antenna pattern modulation, spectrum correction, and integral, and thus, the computational complexity of this method is $o\left(N_{a}N_{r}n_{a}\right)$, which is approximately the same as that of many existing efficient methods. Moreover, memory is used in a zero copy manner [25], which contributes to speeding up the program. Specifically, the complexity does not increase with the increase of deviation magnitude. Thus, this method is particularly suitable for large deviation condition.

### 4.2. Flexibility

This algorithm is compatible with different waveforms, antenna configurations, and multiple acquisition modes.

The method has wide applicability for different waveforms. Note that the transfer function $hh_{tran}\left(\cdot \right)$ is defined in the signal Fourier domain rather than the time domain, which relieves the constraint on the waveform. Thus, the simulator can account for the nonlinear distortion of the transmitted signal, complex waveforms other than the LFM signal, or even waveform agility at different slow-time instants. The cost of such flexibility is not significant. As illustrated in Figure 4, the additional steps are the FFT operation of transfer function, multiplication, and the IFFT operation. They are not very time-consuming compared with the integration.

The method can also account for different antenna configurations. In existing simulators, narrow beam width is often assumed [19,20]. However, the proposed method has relaxed this assumption without sacrificing phase accuracy. As illustrated in (Equation 10), for the wide beam, the integral interval should be increased. Thus, there is a slight reduction in the computational efficiency. In addition, the squint angle is another issue of concern for the SAR simulator. Many efficient algorithms are only suitable for zero or small squint angle [21]. By comparison, our method has a very loose constraint on the squint angle. For different squint angles and beam patterns, we should adjust the integral interval and pattern modulation according to (Equation 10) and (Equation 11). Generally, our approach is compatible with beam width, beam pattern, and squint angle.

The method is also applicable for multiple acquisition modes. There are various SAR acquisition modes for different applications. The typical SAR acquisition modes include stripmap staring spotlight SAR, sliding spotlight SAR, TopSAR, and ScanSAR. The main difference between different models is the method of steering the pointing direction of radar antenna [26]. Figure 2 illustrates some typical acquisition modes. For example, TopSAR and spotlight modes are characterized by the steering of radar antenna to a rotation center. In the TopSAR mode, the rotation center is located in the opposite direction of the illuminated scene, whereas the rotation center of spotlight SAR is located in the same direction of the illuminated scene. Based on the spectrum extraction and antenna pattern modulation operations, this method can account for acquisition modes other than stripmap mode. As the beam pointing direction $\theta_{sq}$ varies with the instantaneous radar location ${\vec{l}}_{r}$, the spectrum extracted should be expanded as described above to reduce the computational complexity because the Stolt interpolation can be applied only once. Furthermore, the window function $\omega \left(\cdot \right)$ should be altered with ${\vec{l}}_{r}$.

### 4.3. Error Analysis and Limitations

The proposed SAR signal simulation method involves three primary approximations: spatial spectrum computation via the Principle of Stationary Phase (POSP), reflectivity map weighting, and slant-range modeling approximation. Each approximation introduces specific constraints, which are analyzed below.

Spatial Spectrum Calculation (POSP): The POSP method is used to approximate the spatial spectrum of the system’s transfer function. It is particularly suitable for quasi-linear frequency-modulated (FM) signals with non-quadratic phase components [24]. Theoretical and experimental analyses show that the maximum phase error introduced by POSP is less than π/10, exhibiting systematic behavior that minimally impacts simulation accuracy.

Reflectivity Map Weighting: The weighting function $v^{-3/2}$ approximates ${\left(v-r\right)}^{-3/2}$ under the assumption that the deviation *r* is much smaller than the map location *v*, and the associated exponential term is negative (see Equations (Equation 35) and (Equation 36)). This approximation introduces negligible phase error, as confirmed by experiment (see Section A.3).

Slant-Range Approximation: The dominant constraint stems from approximating the slant range. A Taylor expansion of Equation (Equation 2) yields(20)$$\Delta R_{0}\approx -d\left(x\right)\cos\theta_{⊥}+\frac{d^{2}\left(x\right)\sin^{2}\theta_{⊥}}{2R_{0}}+\frac{d{\left(x\right)}^{3}\cos\theta_{⊥}\sin^{2}\theta_{⊥}}{2R_{0}^{2}}+\frac{\left(5\cos^{2}\theta_{⊥}-1\right)\sin^{2}\theta_{⊥}}{8R_{0}^{3}}d{\left(x\right)}^{4}$$
with residual error bounded by(21)$$\left|E(R)\right|\leq \left|\frac{\left(5\cos^{2}\theta_{⊥}-1\right)\sin^{2}\theta_{⊥}}{8R_{0}^{3}}d{\left(x\right)}^{4}\right|\leq \left|\frac{d{\left(x\right)}^{4}}{2R_{0}^{3}}\right|$$

To preserve simulation validity, this residual must be much smaller than the radar range resolution:(22)$$\left|\frac{d^{4}}{2R_{0}^{3}}\right|\ll \frac{c}{2B}$$

Here, *d* is the maximum value of trajectory deviation amplitude. It is worth mentioning that higher-order expansions of Equation (Equation 2) could relax constraints on the magnitude of deviation, though practical implementation requires case-specific optimization.

Constraint on Imaging Template Bandwidth: The proposed method imposes a relatively mild constraint on the spatial bandwidth of the imaging template. Since SAR signal bandwidth and Doppler bandwidth are determined by radar system parameters and inversely proportional to resolution, the two-dimensional spatial Fourier transform of the imaging template σ(·) must have sufficient spectral support to cover the full signal bandwidth.

In practice, the spectral content of σ(·) is governed by its spatial sampling interval. A finer grid provides broader spectral coverage, essential to accurately reproduce the frequency content of SAR signals. To prevent spectral truncation or aliasing, the grid spacing should be significantly smaller than the SAR resolution.

Compared to conventional frequency-domain methods that rely on strict linear trajectory assumptions and small deviations, this constraint is relatively loose. Traditional simulators often degrade under large or nonlinear motion. In contrast, the proposed method leverages spatial spectrum modeling to accommodate substantial trajectory deviations while maintaining both high phase fidelity and computational efficiency. This makes it well-suited for dynamic platforms, agile maneuvering, and advanced imaging modes such as staring spotlight, TOPSAR, and sliding spotlight.

Limitation in Interferometric Phase Simulation: It should be emphasized that, consistent with the assumptions of most frequency-domain SAR echo simulators [21,27], the proposed method does not simulate interferometric phase contributions arising from terrain height variations. Specifically, the reflectivity map is defined on a two-dimensional plane, and elevation-induced variations in slant range are not incorporated into the signal model. In practical SAR interferometry, however, these variations result in deterministic phase differences in the interferogram, forming the basis for applications such as digital elevation model (DEM) generation and surface deformation monitoring [28].

This limitation indicates that although the proposed approach can accurately replicate SAR amplitude responses under significant or nonlinear trajectory deviations, it does not support simulation of the interferometric phase. Therefore, the method is not applicable to InSAR scenarios that require phase sensitivity and coherence modeling across multiple acquisitions. Nevertheless, for single-pass SAR imaging and use cases focused on amplitude-domain characteristics—such as resolution validation, motion compensation evaluation, or sensor design—the proposed simulator remains valid, efficient, and accurate. Incorporating elevation-dependent phase terms into the model could serve as a potential direction for extending the method toward interferometric SAR applications.

### 4.4. Application to Bistatic SAR

The proposed method is not limited to conventional monostatic SAR systems but can be naturally extended to bistatic configurations, where the transmitter and receiver are spatially separated. In such cases, the transfer function must account for two distinct geometric paths: from the transmitter to the target, and from the target to the receiver. As shown in Equation (Equation 23), the bistatic transfer function $h_{\mathrm{tran}}$ is expressed as the product of two independent propagation components: (23)$$\begin{array}{cc} hh_{\mathrm{tran}}\left({\vec{l}}_{r_{1}},{\vec{l}}_{r_{2}},{\vec{l}}_{s};k\right) & =s_{\bar{\mathrm{amp}}}\left({\vec{l}}_{r_{1}},{\vec{l}}_{s}\right)\exp\left(-j\pi kR\left({\vec{l}}_{r_{1}},{\vec{l}}_{s}\right)\right)\times s_{\bar{\mathrm{amp}}}\left({\vec{l}}_{r_{2}},{\vec{l}}_{s}\right)\exp\left(-j\pi kR\left({\vec{l}}_{r_{2}},{\vec{l}}_{s}\right)\right)\\ & =hh_{\bar{tran}}\left(\vec{l_{r_{1}}};\vec{l_{s}};k\right)\times hh_{\bar{tran}}\left(\vec{l_{r_{2}}};\vec{l_{s}};k\right)\\ & =hh_{\bar{\mathrm{tran}\_1}}\left(\cdot \right)\times hh_{\bar{\mathrm{tran}\_2}}\left(\cdot \right) \end{array}$$
where $hh_{\bar{\mathrm{tran}\_1}}\left(\cdot \right)$ and $hh_{\bar{\mathrm{tran}\_2}}\left(\cdot \right)$ represent the transmitter-to-target and target-to-receiver propagation terms, respectively. Each term follows the same structure as in the monostatic case, consisting of an amplitude modulation function $s_{\bar{\mathrm{amp}}}\left(\cdot \right)$ and a phase term determined by the slant range $R\left(\cdot \right)$.

A key advantage of this formulation lies in the decoupled structure of the transfer function. This separation allows the proposed two-dimensional (2D) plane-based simulation algorithm to be independently applied to each propagation path. Specifically, both $hh_{\bar{\mathrm{tran}\_1}}\left(\cdot \right)$ and $hh_{\bar{\mathrm{tran}\_2}}\left(\cdot \right)$ can be efficiently computed within their respective 2D spatial domains using the same high-speed computation techniques developed for the monostatic scenario. This modular approach avoids the substantial computational burden typically associated with bistatic SAR simulation, which conventionally requires joint 3D spatial-temporal processing. Furthermore, by isolating the transmitter and receiver geometry, the method enables flexible adaptation to a wide range of bistatic configurations without modification to the core computational framework.

In summary, this extension to bistatic SAR demonstrates the method’s robustness, generality, and scalability. It is particularly advantageous in high-resolution imaging tasks involving spatially diverse transceiver platforms, such as distributed radar networks, or opportunistic bistatic SAR scenarios.

## 5. Simulation Results

### 5.1. Verification of Spatial Spectrum Analysis

The spatial spectrum analysis of the SAR transfer function is the theoretical basis of the proposed method. Thus, simulation is performed to evaluate the spatial spectrum of the SAR transfer function. Assume that the target in consideration is a single scatterer with a constant backscattering coefficient located at ${\vec{l}}_{s}=\left(u,v\right)=\left(0,0\right)$. The radar is a C-band stripmap SAR with the carrier frequency 6 GHz ($k_{0}=40$ $m^{-1}$).

The radar beam pattern is specified as follows:(24)$$\omega \left(\theta ,\theta_{sq}\right)=\cos\left(\pi \frac{\alpha }{\theta_{bw}}\right),\left|\alpha \right|⩽\frac{\theta_{bw}}{2}$$
where the radar squint angle $\theta_{sq}=10\deg$, α is the intersection angle between θ and $\theta_{sq}$, and beam width $\theta_{bw}=5\deg$.

According to the definition of the SAR transfer function given by (Equation 6), we calculate the sampling values of the SAR transfer function in the following region:(25)$$\left\{{\vec{l}}_{r}=\left(x,r\right)\left|-10^{4}\sin\theta_{sq}⩽x⩽0,-10^{4}\cos\theta_{sq}⩽r⩽0\right.\right\}$$

The geometrical settings are illustrated in Figure 6a. The sampling interval of the SAR transfer function is set as Δx=0.0970 m and Δr=0.5501 m. As the distance from the radar to scatterer *R* appears in the denominator of the amplitude of the SAR transfer function as expressed in (Equation 8), it is assumed that the amplitude of the SAR raw signal is zero when R⩽1000 m (i.e., when the radar is within region #2 in Figure 6a, in order to avoid meaningless singular values. Note that the amplitude of the SAR transfer function is also zero within regions #3 and #4 in Figure 6a, because the scatterer would be outside the radar illumination. The amplitude of the simulated SAR transfer function is shown in Figure 6b.

Applying 2D FFT to the samplings of SAR transfer function, we obtain the 2D spatial frequency spectrum of SAR transfer function, as shown in Figure 6c. As predicted using (Equation 10) and Figure 3, it is an arc whose circle radius is exactly equal to the wavenumber $k_{0}$, and the central angle of the arc is equal to the beam width $\theta_{bw}$. The result confirms the spatial spectrum analysis of the SAR raw signal, thus validating the theoretical basis of this method.

### 5.2. Results of Stripmap SAR for Point Scatterer Case

In this section, experiments that simulate the raw signal of stripmap SAR with trajectory deviation are conducted to assess the effectiveness of the proposed method.

The radar parameters are listed in Table 1. As an example, let the target point be located at (1000, 9797.95, −2000) m, with corresponding coordinates in the slant range plane at (0, 10,000) m, so that the radar trajectory is near the origin. Assume that the nominal trajectory coincides with the *x* axis. As shown in Figure 7a, the trajectory deviation is set as(26)$$\left\{\begin{array}{c} y\left(x\right)=100\sin\left(\frac{\pi }{50}x\right)+10rand\\ z\left(x\right)=100\cos\left(\frac{\pi }{50}x\right)+10rand \end{array}\right.$$
where the deviation diameter is set to 200 m. Notably, this deviation is significantly larger than those considered in existing efficient simulators. For instance, the deviation magnitudes are only a few decimeters in [22], several meters in [21], and over ten meters in [27].

First, the SAR raw signal is calculated using the proposed method, as shown in Figure 4. In order to analyze the error of this method, the time-domain method is adapted due to its accuracy. Therefore, the theoretical transfer function is calculated according to its definition in (Equation 6) and (Equation 8). The phase error of the proposed method compared with the theoretical value is illustrated in Figure 7b,c. For all radar locations and all wavenumbers *k* within the bandwidth of the transmitted signal, the mean of absolute phase error is less than 0.0231 rad, the standard deviation of absolute phase error is less than 0.0430 rad, and the maximum of absolute phase error remains below 0.2837 rad. This level of phase deviation corresponds to a path length error of less than λ/10, which is significantly smaller than the system’s resolution in both range and azimuth. As a result, the phase error introduced by the proposed simulation method has negligible impact on SAR image quality. In practical terms, the resulting deviation in simulated echo signals is well within the tolerances of typical SAR imaging systems, and thus, does not degrade the final reconstructed image.

Figure 7b shows that the phase error varies with the radar location when k=39 $m^{-1}$. Notably, when the target point is located near the beam center, the phase error is always less than 0.05 rad. The phase error is over 0.1 rad only when the target point is located at the edge of the radar beam. Moreover, the amplitude of this part of the SAR raw signal is much less than that of others due to the radar beam pattern, thus causing a slight effect on the SAR image formation processing. This result aligns with the analysis in Section 4.3 and Section A.3, where we established that the primary source of phase error is the approximation introduced by the principle of stationary phase (POSP). This error is systematic and remains below π/10, which corresponds to a path length deviation smaller than λ/10. Such an error level is well within acceptable limits for SAR imaging and has negligible impact on image quality. This is also one of the key reasons why the POSP method is widely applied in existing efficient simulators [19,20]. In Figure 7c, it is shown that the magnitude of phase error is independent of the frequency of the transmitted radar signal. Thus, the proposed simulator is comparable to a radar signal with large bandwidth. In summary, the simulation results indicate that the proposed SAR raw signal simulator is applicable for the trajectory deviation condition.

In order to accurately verify the performance of the algorithm, we carried out imaging processing on the echo of the above point target. After motion compensation, the ω−k imaging algorithm was applied for image formation and upsampling. The resulting local radar images, azimuth profiles, and range profiles of the point scatterers are shown in Figure 8. As observed, the point targets are well-focused, with Figure 8 b,c demonstrating complete focus and ideal Sinc-pattern sidelobes. The range-direction mainlobe width was measured at 0.44 m, consistent with the radar’s theoretical resolution, and the peak sidelobe ratio (PSLR) was −13.3 dB. Due to the directional characteristics of the radar antenna, the azimuth-direction sidelobes were lower than those in the range direction, with a mainlobe width of 0.45 m and a PSLR of −13.4 dB. The experimental results confirm that the proposed method is effective for simulating SAR raw signals under trajectory deviation conditions, thereby validating its applicability.

### 5.3. Results of Stripmap SAR for General Electromagnetic Case

To further validate the effectiveness of the proposed SAR signal simulation method in real-world radar scenarios, we conducted a simulation experiment using a realistic scene. Echo data were generated following the procedure, as illustrated in Figure 4. The deviation model employed is consistent with Equation (Equation 26), introducing substantial deviations from the nominal flight path. For evaluation, a reference image was selected from a TerraSAR-X staring spotlight mode acquisition over the Aswan Dam in Egypt. The simulated scene was configured with a resolution of 0.1 m, consisting of 7500 pixels in range and 6000 pixels in azimuth. Moreover, in order to more comprehensively assess the performance of the simulator, an additional set of echo data was generated without applying any trajectory deviation. Both datasets were processed through image reconstruction. Notably, due to the large trajectory offsets, the azimuth chirp rate of the signal became highly distorted. To address this, we adopted an azimuth sub-aperture imaging approach and motion compensation. The resulting SAR images are shown in the left column of Figure 9.

Compared with the reference template shown in Figure 9a, both images—generated with and without trajectory deviation—preserve the key structural characteristics of the scene. Critical features such as the dam, surrounding water bodies, and urban infrastructure are clearly identifiable, and the overall image quality remains high. To provide a more detailed evaluation, specific regions of interest within the images were magnified, as presented in the right column of Figure 9. While minor defocusing is observed compared to the original template (which has a 0.1 m resolution), the reconstructed SAR images are consistent with the expected 0.5 m radar resolution. Notably, stronger defocusing effects are observed in both range and azimuth directions in the case with trajectory deviation. This is attributed to the use of first-order motion compensation only. Nonetheless, the visual differences are minimal, and the overall impact of the trajectory deviation on image quality is negligible.

In summary, the results confirm the accuracy and robustness of the proposed SAR signal simulation method under large-scale trajectory deviations. The successful reconstruction of high-quality imagery under both ideal and perturbed flight conditions demonstrates the method’s suitability for advanced SAR applications involving complex or nonlinear platform motion.

### 5.4. Results of Spotlight SAR with Trajectory Deviation

In this section, the spotlight mode is considered as an example of SAR acquisition mode in which the squint angle $\theta_{sq}$ varies with slow time. For the spotlight mode, the rotation center is located in the same direction as the illuminated scene. We assume the same radar parameters as those listed in Table 2. Note that this simulation is an example of a very high resolution radar with bandwidth as much as 1 GHz and range resolution of approximately 0.15 m. Let the trajectory deviation be a discontinuous function with random deviation, which is the same as (Equation 26). The target point is assumed at $\left(0,\;9797.95,\;-2000\right)$ m, with corresponding coordinates in the slant range plane being (0, 10,000) m. We calculate the transfer function $hh_{trans}\left(\cdot \right)$ using the proposed method and time-domain method separately.

The simulation results are shown in Figure 10. As the squint angle of radar varies along the trajectory, the phase error is plotted as a function of $\theta -\theta_{sq}$ at *k* = 39 m^−1^ in Figure 10a. We can observe that the result is similar to the experiment with stripmap SAR in Figure 7b. The phase error is over 0.05 rad only when the target point is located at the edge of the radar beam. In order to assess the performance of the proposed method for wideband transmitted radar signals, the phase error is viewed as a function of $\theta -\theta_{sq}$ and transmitted signal frequency *k*, as shown in Figure 10b. Similar to Figure 7c, the magnitude of the phase error remains nearly constant across the bandwidth at any fixed radar location. For all combinations of *k* and radar positions, the mean absolute phase error does not exceed 0.0105 rad, the standard deviation remains below 0.0276 rad, and the maximum absolute phase error is under 0.2065 rad. This maximum phase error is well below π/10, corresponding to a path deviation of less than λ/10, which is significantly smaller than the resolution of the SAR system. Hence, the phase error introduced by the proposed simulator can be regarded as negligible for SAR image formation.

In conclusion, the effectiveness of the proposed method for spotlight mode is validated. Furthermore, the simulator performs reliably even under large signal bandwidths. Given that spotlight mode involves a time-varying squint angle $\theta_{sq}$, these results demonstrate that the simulator is broadly applicable across various SAR acquisition modes.

## 6. Conclusions

This paper presents a new methodology for simulating the raw signal of SAR with large trajectory deviation. This approach is based on the spatial spectrum analysis of the SAR raw signal, which views this signal as a function of radar location at a fixed frequency of the transmitted radar signal. By revealing that the SAR raw signal is an arc in the 2D spatial spectrum domain, the proposed simulator calculates the SAR raw signal using a 1D curvilinear integral in the 2D spectrum.

Theoretical analysis and simulation results confirm that the proposed method can accurately simulate raw signals under substantial motion deviations. Compared to conventional frequency-domain simulators, it substantially relaxes constraints on trajectory linearity, deviation magnitude, squint angle, beam width, and signal bandwidth. Moreover, it maintains computational efficiency regardless of deviation magnitude, making it especially suitable for dynamic or maneuvering platforms.

The proposed method demonstrates strong flexibility and generality, supporting a wide range of radar waveforms and SAR acquisition modes, including stripmap, spotlight, TopSAR and sliding spotlight. Moreover, its characteristics of spatial spectrum and compatibility with nonlinear trajectory make it particularly suitable for advanced scenarios such as deception jamming, bistatic SAR, and circular SAR (CSAR), where high phase fidelity and flexible geometric modeling are essential.

In conclusion, this work provides a unified, accurate, and computationally efficient framework for SAR raw signal simulation under large trajectory deviations. Its robustness across different configurations and motion patterns makes it a valuable tool for both traditional SAR system design and advanced SAR applications under complex motions and multi-platform configurations. 

## Figures and Tables

**Figure 1 sensors-25-04260-f001:**
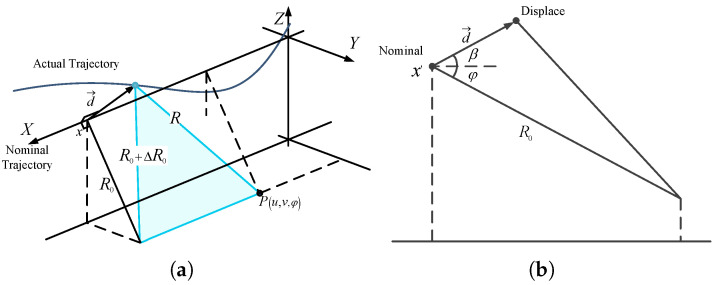
Geometric model of SAR: (**a**) with trajectory deviation; (**b**) inCrossTrackPlane.

**Figure 2 sensors-25-04260-f002:**
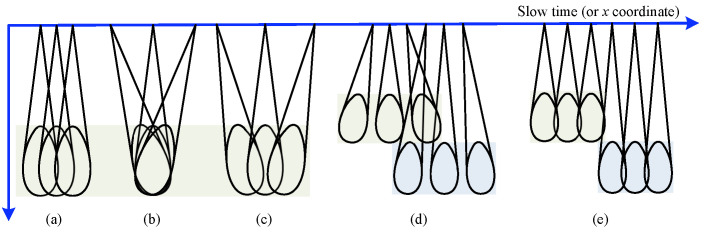
Illustration of the antenna steering in typical SAR acquisition modes including: (**a**) stripmap; (**b**) staring spotlight; (**c**) sliding spotlight; (**d**) TopSAR; (**e**) ScanSAR. Different colors represent different detection ranges.

**Figure 3 sensors-25-04260-f003:**
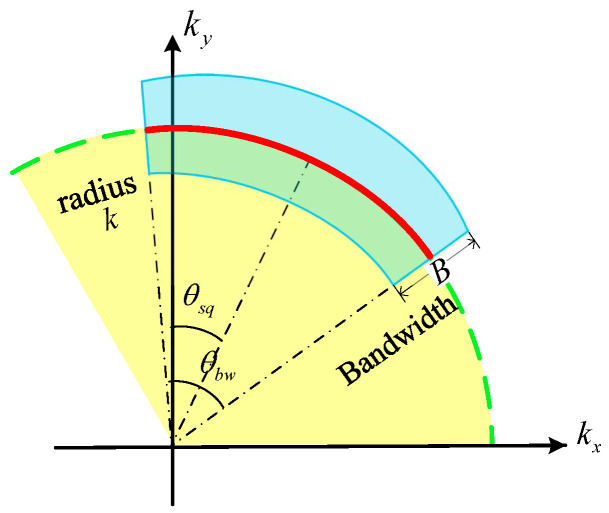
Two-dimensional frequency spectrum of SAR raw signal (transfer function), as an arc whose radius is *k* and central angle is $\theta_{sq}$. The blue-shaded region denotes the frequency spectrum range of the radar signal.

**Figure 4 sensors-25-04260-f004:**
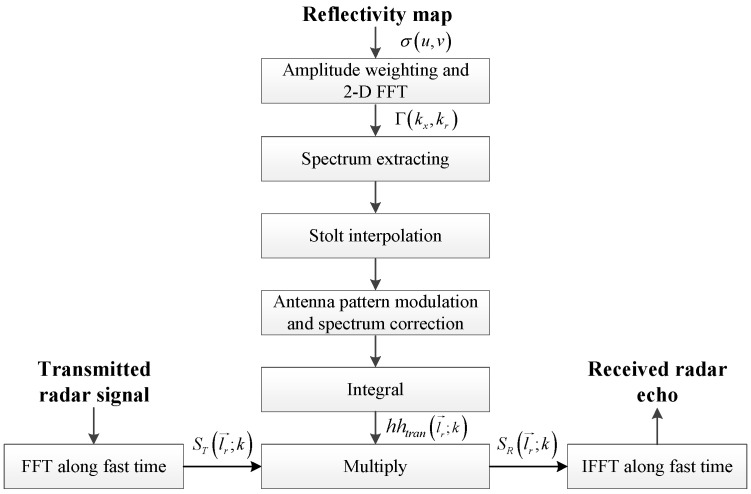
Block scheme of the proposed simulation approach.

**Figure 5 sensors-25-04260-f005:**
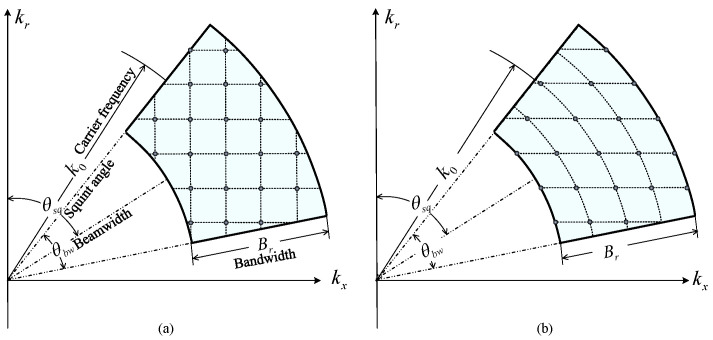
Sampling grid and support region of: (**a**) $\Gamma \left(k_{x},k_{r}\right)$ before Stolt interpolation in coordinates $\left(k_{x},k_{r}\right)$; (**b**) $\Gamma \left(k_{x},k_{r}\right)$ after Stolt interpolation in coordinates $\left(k_{x},k_{r}\right)$.

**Figure 6 sensors-25-04260-f006:**
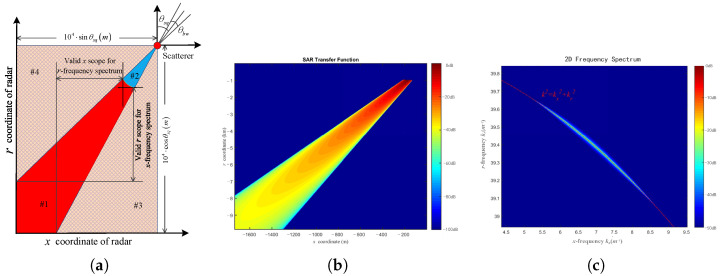
SAR transfer function: (**a**) geometry setting; (**b**) 2D spatial domain; (**c**) 2D spatial frequency spectrum.

**Figure 7 sensors-25-04260-f007:**
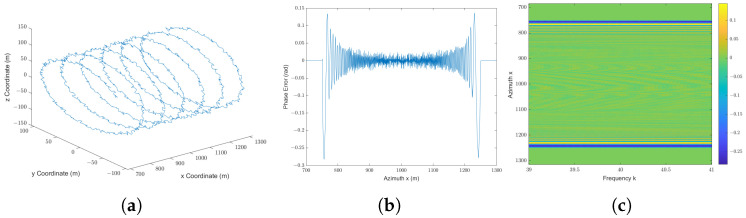
Simulation results for stripmap mode with trajectory deviation: (**a**) SAR trajectory with deviation; (**b**) phase error plotted as a function of radar *x* coordinate at *k* = 39 $m^{-1}$; (**c**) phase error plotted as a 2D function of transmitted signal frequency *k* and radar *x* coordinate.

**Figure 8 sensors-25-04260-f008:**
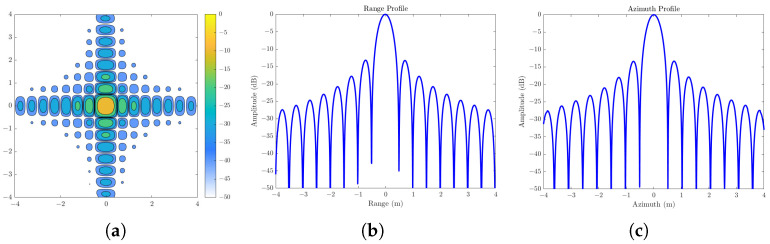
Image formation processing results of stripmap SAR raw signal of point scatterer: (**a**) close-up image of image formation processing; (**b**) profile of range; (**c**) profile of azimuth.

**Figure 9 sensors-25-04260-f009:**
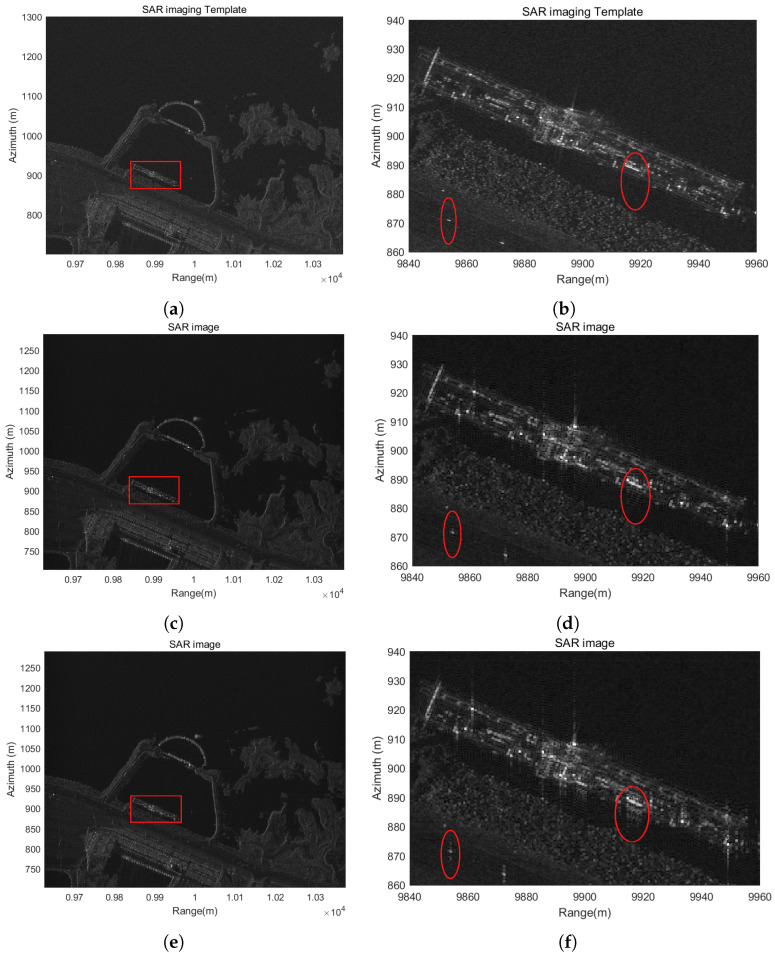
Reconstructed SAR images under different trajectory conditions using the proposed simulation method: (**a**) reference TerraSAR-X image used as the imaging template; (**b**) magnified view of the selected region from (**a**); (**c**) reconstructed image without trajectory deviation; (**d**) magnified view of the selected region from (**c**); (**e**) reconstructed image with trajectory deviation; (**f**) magnified view of the selected region from (**e**). The red squares denote magnified regions of interest, while the red ovals mark representative sample areas analyzed for defocusing effects.

**Figure 10 sensors-25-04260-f010:**
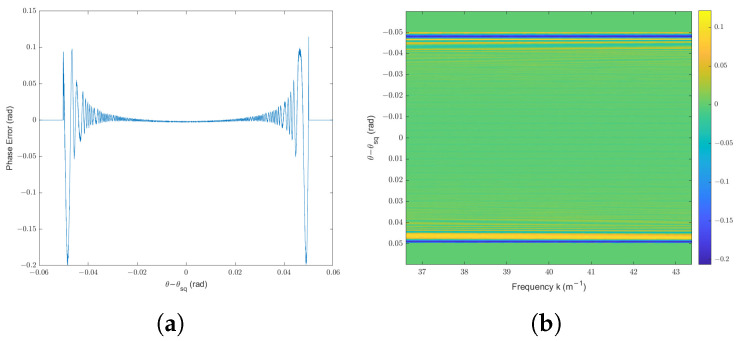
Simulation results for the spotlight mode with trajectory deviation: (**a**) phase error plotted as a function of $\theta -\theta_{sq}$ at *k* = 39 m^−1^; (**b**) phase error plotted as a 2D function of the transmitted signal frequency *k* and $\theta -\theta_{sq}$.

**Table 1 sensors-25-04260-t001:** Parameter settings of stripmap SAR.

Parameter Type	Value
resolution	≈0.5 m (both range and azimuth)
carrier frequency	6.0 GHz, $k_{0}$ = 40 $m^{-1}$
bandwidth	$B_{w}$ = 300 MHz, 2Bw2Bwcc=2 $m^{-1}$
squint angle	$\theta_{sq}$ = 0 rad
azimuth beam width	$\theta_{bw}$ = 0.05 rad
azimuth beam pattern	$\omega \left(\alpha \right)=rect(\alpha /\theta_{bw})$,
	where α is the angle between θ and $\theta_{\mathrm{sq}}$

**Table 2 sensors-25-04260-t002:** Parameter settings of spotlight SAR.

Parameter Type	Value
resolution	≈0.15 m (range)
carrier frequency	6.0 GHz, $k_{0}$ = 40 m^−1^
bandwidth	$B_{w}$ = 1 GHz, 2Bw2Bwcc = 6.67 m^−1^
antenna rotation center	(0, 15) km
azimuth beam width	$\theta_{bw}$ = 0.05 rad
azimuth beam pattern	$\omega \left(\alpha \right)=e^{-2{\left(\frac{2\alpha }{\theta_{bw}}\right)}^{2}}$,
	where α is the angle between θ and $\theta_{sq}$

## Data Availability

The data presented in this study are available on request from the corresponding author due to the relevant requirements of the railway company.

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
