# Peer review of "An Efficient SAR Raw Signal Simulator Accounting for Large Trajectory Deviation"

_sensors, 2025, doi:10.3390/s25144260_

Round 1
Reviewer 1 Report
Comments and Suggestions for Authors
This manuscript presents a novel SAR raw signal simulator based on spatial spectrum analysis, demonstrating capabilities for handling large trajectory deviations while maintaining high computational efficiency and broad applicability. The following constructive suggestions could further enhance the quality of the work:
1. Redundant notation ("where α") appears in Table 2 and should be revised for consistency.
2. Formatting issues require attention:
- The indentation of "where" in line 120 should be removed.
- The capitalized "Where" in line 519 should be corrected to lowercase.
3. The phase error analysis in Figures 7 and 10 would benefit from:
- Explicit discussion of error sources and their quantitative impact on imaging quality.
- Comparative experiments to validate the real-scene imaging results in Figure 9 (e.g., against measured data or established simulation methods).
4. The approximation conditions in Equations (2) and (3) require clearer delineation of their validity boundaries and operational constraints.
5. The introduction of the curvilinear integral in Equation (9) lacks contextual motivation. A physical interpretation of this mathematical formulation should be provided to improve conceptual clarity.
6. The conclusion section could be strengthened by incorporating a forward-looking perspective on potential extensions or applications of the proposed method.
Author Response
1.Redundant notation ("where α") appears in Table 2 and should be revised for consistency.
Response:
We sincerely appreciate the reviewer’s careful attention to detail. The redundant notation "where α" in Table 2 has been removed.
2.Formatting issues require attention:
- The indentation of "where" in line 120 should be removed.
- The capitalized "Where" in line 519 should be corrected to lowercase.
Response:
Thank you for highlighting these formatting inconsistencies. We have carefully revised the manuscript as follows:
The indentation of "where" in line 120 has been removed.
The capitalized "Where" in line 519 (now addressed in line 586) has been corrected to lowercase.
- The phase error analysis in Figures 7 and 10 would benefit from:
- Explicit discussion of error sources and their quantitative impact on imaging quality.
- Comparative experiments to validate the real-scene imaging results in Figure 9 (e.g., against measured data or established simulation methods).
Response to Comment on Phase Error Analysis (Figures 7 and 10):
We thank the reviewer for the constructive suggestion regarding the phase error analysis. In response, we have enhanced our discussion to explicitly identify that the phase errors in Figures 7 and 10 primarily originate from the Principle of Stationary Phase (POSP) approximation during spectrum derivation, with maximum magnitudes below radians (corresponding to path length errors smaller than ). This level of error has been demonstrated to have negligible impact on SAR imaging quality, as thoroughly analyzed in Section 4.3 "Error Analysis and Limitations" and Appendix 7.3 "The Simulation of the Error Analysis". The comprehensive analysis confirms these phase errors are significantly smaller than the system's resolution capability, maintaining the method's accuracy for practical SAR applications.
Response to Comment on Figure 9:
Thank you for your valuable suggestion. In our work, the echo signals are simulated with a phase error smaller than , which corresponds to a range error less than . Such a small range error leads to a negligible effect on the SAR imaging results, as has already been validated through point target imaging experiments. Therefore, the time-domain simulation method used in real-scene imaging (as shown in Figure 9) can be considered to produce results that are almost identical to the actual imaging outcome. Moreover, existing simulation methods that might be used for comparison are typically not applicable under the large cross-track deviations considered in our work. Thus, adding a comparative figure would not provide additional value or insight. Based on these considerations, we respectfully suggest not adding further comparative figures.
- The approximation conditions in Equations (2) and (3) require clearer delineation of their validity boundaries and operational constraints.
Response:
We thank the reviewer for this valuable suggestion. In response, we have added a detailed analysis of the validity boundaries and operational constraints for the approximations in Equations (2) and (3) within Section 4.3 ("Error Analysis and Limitations"). Specifically, Equation (22) in the revised manuscript rigorously quantifies the conditions under which these approximations hold, including explicit criteria for trajectory deviation magnitude and detection range.
- The introduction of the curvilinear integral in Equation (9) lacks contextual motivation. A physical interpretation of this mathematical formulation should be provided to improve conceptual clarity.
Response:
We deeply appreciate this suggestion. Section 3.1 ("Mathematical Principle") has been revised to clarify the physical motivation behind the curvilinear integral in Equation (9),
linking it to the spatial spectrum analysis framework. (See in line 159 to169)
- The conclusion section could be strengthened by incorporating a forward-looking perspective on potential extensions or applications of the proposed method.
Response:
Thank you for emphasizing the value of broader implications. The Conclusion section now includes potential extensions of the method, such as applications to bistatic SAR, circular SAR, and SAR deception jamming scenarios. (See in line 550 to 555)

Reviewer 2 Report
Comments and Suggestions for Authors
The paper proposes a new SAR raw signal simulator, based on the superposition, in the 2D spectral domain, of the returns as seen from a non-rectilinear trajectory of the transmitter. The efficiency comes from the fact that, apparently neglecting 3D effects, the 2D spectra depend only on the antenna direction and on the instantaneous transmitter location and can be superposed efficiently. The results for a point scatterer and for a SAR image of the Aswan dam are shown with good results.
The paper is indeed interesting and it is my opinion that it should be published. However, no mention is made of a possible site topography and its impact on the image and on the simulation. A joint analysis of the limits on topography and trajectory deviations and say the possible results on towns or on hilly terrains should be added, or maybe postponed if it proves to be too difficult, but indeed should be considered. The phase histories of two point-scatterers at different height but at the same range (say at broadside) would depend on the trajectory if not rectilinear, but maybe not too much. The slant planes (Eq. 3, 4) would not coincide, I believe. For example, the simulation in Fig. 7 should be also repeated for points with a height difference say of 10, 30, 100 m.
As said, a joint limit on the topography and the acceptable trajectory deviation should then be given.
Further, the authors should consider the case of bistatic SAR and discuss the possibilities of extending the simulation to that case too.
I think that both points should be considered to help the readers and therefore it is my opinion that the paper should undergo a major revision considering:
- The impact of terrain topography, from urban areas [a] to hilly terrains; to do that, the 3D effects should be considered from the beginning and the Appendix should be corrected consequently. The simulation should be extended to layovers.
- The possibility of extension to bistatic SAR
[a] G. Franceschetti, A. Iodice, D. Riccio and G. Ruello, "SAR raw signal simulation for urban structures," in IEEE Transactions on Geoscience and Remote Sensing, vol. 41, no. 9, pp. 1986-1995, Sept. 2003
Author Response
Reviewer Comment:
The paper is indeed interesting and it is my opinion that it should be published. However, no mention is made of a possible site topography and its impact on the image and on the simulation. A joint analysis of the limits on topography and trajectory deviations and say the possible results on towns or on hilly terrains should be added, or maybe postponed if it proves to be too difficult, but indeed should be considered... Further, the authors should consider the case of bistatic SAR and discuss the possibilities of extending the simulation to that case too.
Response:
We sincerely thank the reviewer for the valuable and insightful comments.
1、On topography and trajectory deviation limits:
We fully agree with the reviewer that the impact of topography, especially in urban or hilly terrain, is an important aspect to consider in SAR raw signal simulation. While the current work focuses on a 2D spectral-domain method, we have conducted a detailed analysis of the trajectory deviation constraints in Section 4.3: Error Analysis and Limitations, which provides quantitative bounds on trajectory deviations to ensure imaging accuracy.
We acknowledge that incorporating 3D topographic effects (e.g., point scatterers at different elevations) would further improve the realism and applicability of the simulation. However, this extension involves additional complexity beyond the scope of the current study and will be pursued in our future work.
2、On the extension to bistatic SAR:
We appreciate the reviewer’s suggestion to explore the bistatic SAR case. In response, we have added a new subsection, Section 4.4: Application to Bistatic SAR, where we discuss the feasibility of extending our simulation framework to bistatic configurations.
We thank the reviewer again for these constructive suggestions, which have helped us to improve and clarify the scope and potential of our work.
Round 2
Reviewer 2 Report
Comments and Suggestions for Authors
I am sorry, but we have to clarify this point before publication.

Author Response
We sincerely thank the reviewer for the careful reading and insightful suggestion. Your attention to detail and rigorous evaluation are greatly appreciated and have significantly contributed to improving the quality of our manuscript.
Comment:
Comparative experiments to validate the real-scene imaging results in Figure 9 (e.g., against measured data or established simulation methods).
Response:
Thank you for this valuable comment. To address this point, we have added a real-scene comparative validation in the revised manuscript. Specifically, we selected a TerraSAR-X image over the Aswan Dam in Egypt as the reference template and used it to validate the simulation results. Echo data were simulated under both ideal and large-scale trajectory deviation conditions using the proposed method. After applying motion compensation and sub-aperture imaging, the reconstructed SAR images are presented in Fig. 9(c)–(f). These results demonstrate that the key structural features are well preserved, and although minor defocusing appears under large deviations, the overall image quality remains high and consistent with expected SAR resolution. This comparison verifies the accuracy and applicability of our simulation method under realistic and challenging conditions.
We hope this addresses your concern satisfactorily. If there are any further questions or suggestions, please do not hesitate to let us know. Thank you again for your time and constructive feedback.
Round 3
Reviewer 2 Report
Comments and Suggestions for Authors
Please see the attachment
